# Effective Upgrading of Levulinic Acid into Hexyl Levulinate Using AlCl₃·6H₂O as a Catalyst

**Valeria D'Ambrosio [1,2,*] and Carlo Pastore [1]**

[1] Water Research Institute (IRSA), National Research Council (CNR), Via F. de Blasio 5, 70132 Bari, Italy; carlo.pastore@ba.irsa.cnr.it

[2] Department of Chemistry, University of Bari Aldo Moro, Via Orabona 4, 70126 Bari, Italy

[*] Correspondence: valeria.dambrosio@ba.irsa.cnr.it; Tel.: +39-0805820505

**Abstract:** AlCl₃·6H₂O was used as a catalyst in the esterification reaction of levulinic acid with 1-hexanol for producing hexyl levulinate, a compound that finds applications in several industrial sectors and represents an excellent candidate to be used in diesel fuel blends. A kinetic and thermodynamic study of the esterification reaction was performed, considering four different temperatures (338, 348, 358, and 368 K), an acid: alcohol: catalyst 1:1:0.01 molar ratio, and a reaction time of 72 h. An optimization study was then carried out, evaluating the effect of alcohol and catalyst amounts, and, in the best reaction conditions (acid:alcohol:catalyst 1:2:0.1), a very high levulinic acid conversion (92.5%) was achieved. By using AlCl₃·6H₂O, alongside the high reaction yield, the product purification was also simplified, being such a catalyst able to trap most of the water in a different phase than hexyl levulinate, and, furthermore, it was found to be completely recoverable and reusable for several reaction cycles, without losing its catalytic effectiveness. The use of AlCl₃·6H₂O, therefore, represents a promising effective green route for obtaining hexyl levulinate.

**Keywords:** hexyl levulinate; direct esterification; AlCl₃·6H₂O; catalyst recoverability; process intensification





## 1. Introduction

Climate change and all the disastrous consequences deriving from it represent a concrete and very urgent problem, and a sudden change in our society is necessary to face it [1,2]. Research efforts are therefore directed at ending the reliance of energy and industrial sectors on fossil fuels, driving a transition towards bio-energy and bio-refinery where alternative renewable raw materials play a protagonist role [3,4]. Biomass, since its large amounts and diverse composition, has an inestimable potential to contribute as cost-effective renewable resources for the development of bioenergy and bio-based platform molecules, from which a wide range of bio-products can be obtained [5–7]. This concept can be summed up in one word, namely bio-refining, which is a concept similar to the petrochemical refinery, but from which it differs in terms of the raw materials used, encompassing all the possible different bio-chemical refining processes starting from renewable organic materials. Levulinic acid plays a key role in the biorefinery context. Also known as 4-oxopentanoic acid, levulinic acid can be obtained from lignocellulosic biomass [8,9], and it represents a very interesting bio-based platform because of its versatile chemistry, including one carbonyl, one carboxyl, and α-H in its inner structure [10], allowing it to be converted to several chemicals or materials [11,12]. Indeed, it has been claimed to be one of the promising top 12 bio-based building blocks for the chemical industry by the United States Department of Energy [11], and it can be exploited for obtaining fuel additives, solvents, flavor substances, pharmaceutical agents, coating, dyes, rubber and plastic additives, and other materials [13].

Among levulinic acid derivates, alkyl levulinates, obtainable from the esterification of levulinic acid with different alcohols, can be used in blends with diesel fuel, improving the

stability of ignition, viscosity, density, lubrication, and cold-flow properties [14–17]. Particularly, levulinate esters with medium-long chains, like hexyl levulinate, have more suitable properties for such use: the long carbon chain provides a higher carbon content and strong hydrophobicity, leading to improved energy density and water insolubility [11]. Furthermore, it has been observed that the addition of such compounds into fuel leads to cleaner combustion processes with a reduction in emissions of hazardous pollutants like nitrogen oxides ($NO_x$) and particulate matter (PM) [17,18]. Alongside its use as a fuel additive, hexyl levulinate can be used as a flavoring agent, fragrance, plasticizer, solvent, and intermediate for developing new chemicals [18]. Esters are usually obtained by Fisher esterification, commonly carried out by using mineral acids as homogeneous catalysts, such as sulfuric and hydrochloric acid, which have proven to be very effective in the catalysis of esterification reaction [8], but have to face problems of reusability and liquid waste disposal [19,20]. To overcome such issues, up to now, several functionalized materials have also been tested as heterogeneous catalysts for the esterification reaction of levulinic acid with several alcohols: zeolites [21,22], acid ion-exchange resins [23], nano-structured solid acids [24], sulfonic acid-functionalized organic polymers [25], silica-based [26], nano-material catalysts [27], ionic liquid [28], and biocatalyst [29] were very effective for obtaining alkyl levulinates. Nevertheless, high temperatures and long reaction times, or both, are typically needed for achieving high yields, and it must also be considered that the preparation of heterogeneous catalysts is often complicated, leading to a consumption of time and money [11].

Recently, aluminum chloride hexahydrate ($AlCl_3 \cdot 6H_2O$) resulted to be very efficient in catalyzing the esterification reaction under homogeneous catalysis and useful for simplifying the purification of the product, inducing a favorable separation of the resulting esters by the co-produced water, in two different phases [30,31]. In detail, at the end of esterification reactions catalyzed by $AlCl_3 \cdot 6H_2O$, the catalyst was observed to be completely solubilized in the bottom phase, where almost all of the water produced by the reaction (>90% wt) was present, and the ester completely present in the top phase, together with the unreacted reagents. Furthermore, it was completely recoverable and reusable for several reaction cycles [31] and, according to an LCA study for biodiesel production, it was found to be the less impacting catalyst compared to sulfuric acid and some solid acid catalysts (mixed metal oxides, functionalized halloysites, mesoporous perovskite, and functionalized silicas) [32]. In this work, the use of $AlCl_3 \cdot 6H_2O$ for the esterification reactions of levulinic acid with 1-hexanol was investigated. Kinetics ($k_f$, $k_r$, $E_a$, $E_a^{-1}$) and thermodynamic parameters ($K_{eq}$, $\Delta H^0$, $\Delta S^0$) were determined. Moreover, the effect of the amount of $AlCl_3 \cdot 6H_2O$ and 1-hexanol on the reaction yield, the relationship between $K_{eq}$ and the catalyst loading, and the recoverability and reusability of the catalyst were investigated.

## 2. Materials and Methods

### 2.1. Materials

Aluminum chloride hexahydrate ($AlCl_3 \cdot 6H_2O$, 99%) was purchased from Baker (Phillipsburg, NJ, USA); levulinic acid ($CH_3CO(CH_2)_2COOH$, ≥98%) and 1-hexanol ($CH_3(CH_2)_5OH$, >98%) were purchased from Acros Organics (Waltham, MA, USA). Phenolphthalein (99%), potassium dichromate ($K_2Cr_2O_7$, 99%), EDTA disodium salt (99%), 0.1 N KOH solution, and 0.1 N $AgNO_3$ solution were Sigma-Aldrich products (St. Louis, MO, USA).

### 2.2. Determination of Reactants, Products, and $AlCl_3 \cdot 6H_2O$

Characterization of reactants, products, and $AlCl_3 \cdot 6H_2O$ was carried out by performing GC-FID and GC-TCD analysis and acid–base, argentometric, and complexometric titrations, using GC-FID and GC-TCD methods and other procedures previously reported in another study [31]. Injections were performed in splitless mode on an Agilent 8890 GC-FID equipped with a DB-FATWAX UI (30 m, 0.25 mm, 0.25 μm, Agilent Technologies, Santa Clara, CA, USA), with an injector temperature of 523 K, using helium as a carrier gas, with a flow of 2.8 mL·min$^{-1}$. The temperature was initially set to 333 K, holding for

4 min; then, it was increased to 473 K (rate of increase 10 K·min$^{-1}$), holding for 1 min, up to a final temperature of 513 K (rate of increase of 20 K·min$^{-1}$), held for 3 min. The temperature of the detector (FID) was set to 613 K. Water was determined on an Agilent 8890 GC-TCD equipped with an HP-5MS capillary column (30 m; Ø 0.32 mm; 0.25 μm, Agilent Technologies). Injections were performed using a 25:1 split ratio, an injector temperature of 523 K, and helium as a carrier gas, with a flow of 28 mL·min$^{-1}$. The initial oven temperature was set to 313 K, held for 2 min; then, the temperature was increased to 523 K (rate of increase 35 K·min$^{-1}$), held for 5 min. Acid–base titrations were performed using a sample amount of 0.3 g, a 0.1 N KOH aqueous solution, and phenolphthalein as an indicator. AlCl$_3$·6H$_2$O content was determined by titrating Al$^{3+}$ and Cl$^-$. In detail, Cl$^-$ determination was performed through a precipitate titration using a 0.1 N AgNO$_3$ solution, while Al$^{3+}$ was determined through a complexometric back-titration using an excess of a 0.02 N EDTA aqueous solution, back-titrated with a 0.02 N Mg$^{2+}$ aqueous solution [31].

### 2.3. Esterification Reaction of Levulinic Acid with 1-Hexanol Using AlCl$_3$·6H$_2$O as Catalyst

Esterification reaction of levulinic acid with 1-hexanol was performed in glass reactors equipped with silicone caps, allowing the sampling throughout the reaction. Reactions were performed at four different temperatures, namely 338, 348, 358, and 368 K, using a levulinic acid: 1-hexanol molar ratio (R) equal to 1 or 2 and different catalyst loadings, equal to 1–10% mol with respect to the acid.

Aliquots of 0.3 mL were sampled at different times, and the composition was determined by titrations (acid–base, argentometric, and complexometric titrations) and GC-TCD and GC-FID analysis. A biphasic system was observed at the end of each esterification reaction: the two phases were physically separated by using a glass pipette, weighed in two different vials, and their composition was determined. In particular, top phase composition was determined by performing GC-TCD and GC-FID analysis: reactants (levulinic acid and 1-hexanol) and products (hexyl levulinate and water) were determined. As for the bottom phases, being aqueous acid solutions, a very small amount of such phase was considered, and it was diluted with 1-butanol (dil 1:100) and injected into GC-TCD for evaluating any trace of reactants and products. For levulinic acid determination in the lower phase, an acid–base titration was also performed, subtracting the contribution of the acidic catalyst to the consumption of the base. All the experiments were performed in triplicate, observing a standard deviation below 5%.

## 3. Results

### 3.1. Kinetic and Thermodynamic Study of the Esterification Reaction of Levulinic Acid with 1-Hexanol

The kinetic and thermodynamic study was carried out considering four different temperatures (338, 348, 358, and 368 K), an acid: alcohol: catalyst 1:1:0.01 molar ratio, and a reaction time of 72 h. The esterification reaction was the only occurring reaction, observing only water and hexyl levulinate as products, confirming the absence of side-products. The conversion of levulinic acid at the four different temperatures was monitored over time (Figure 1), and for each temperature, after 72 h, no changes were observed in the conversion, confirming the equilibrium was reached.

The equilibrium constant ($K_{eq}$) was thus calculated considering the conversion at the equilibrium $X_{eq}$ (Equation (1)), and rate constants of forward reactions ($k_f$) at the four different temperatures were determined by plotting the equation specified in Banchero's study (Equation (2)) for a second-order homogeneous reaction, valid when an alcohol: acid molar ratio equal to one is used [19,33].

$$K_{eq} = \frac{X_{eq}^2}{\left(1 - X_{eq}\right)^2} \tag{1}$$

$$\frac{\ln Y}{a_2} = \frac{1}{a_2} \ln\left[\left(\frac{2a_1 X_A - 1 - m - a_2}{2a_1 X_A - 1 - m + a_2}\right)\left(\frac{-1 - m + a_2}{-1 - m - a_2}\right)\right] = k_f t \tag{2}$$

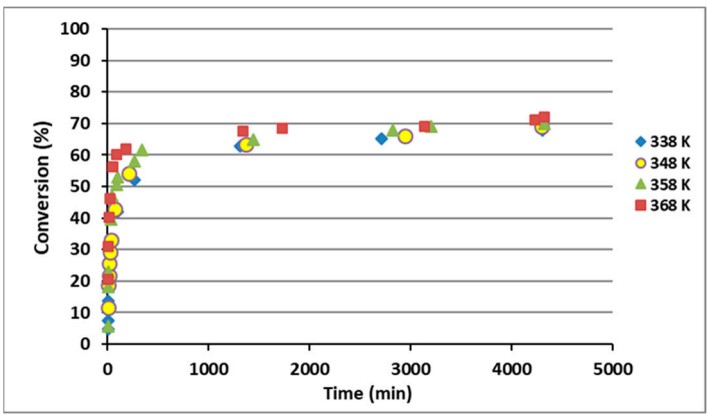

**Figure 1.** Conversion of levulinic acid determined at different temperatures (338–368 K) and time (0–72 h).

With $a_1 = (1 - 1/K_{eq})$; $a_2 = [(m + 1)^2 - 4a_1 \cdot m]^{1/2}$; and m = alcohol/acid molar ratio.

Considering the relationship between $k_f$ and $K_{eq}$, the rate constant of the reverse reactions $k_r$ was calculated (Equation (3)) as well as the activation energies of the forward and reverse reactions ($E_a$, $E_a^{-1}$), and thermodynamic parameters ($\Delta H^0$, $\Delta S^0$) considering Arrhenius and Van 't Hoff equations, respectively (Equations (4) and (5)) [31].

$$k_r = \frac{k_f}{K_{eq}} \tag{3}$$

$$\ln(k'_1) = \ln(A) - \frac{E_a}{R \cdot T} \tag{4}$$

$$\ln(K_{eq}) = -\frac{\Delta H^0}{R \cdot T} + \frac{\Delta S^0}{R} \tag{5}$$

Figure 2a shows the plotting of Equation (2), allowing the determination of $k_f$, while Figure 2b Arrhenius and Van 't Hoff equations for the determination of $E_a$, $E_a^{-1}$, $\Delta H^0$, $\Delta S^0$.

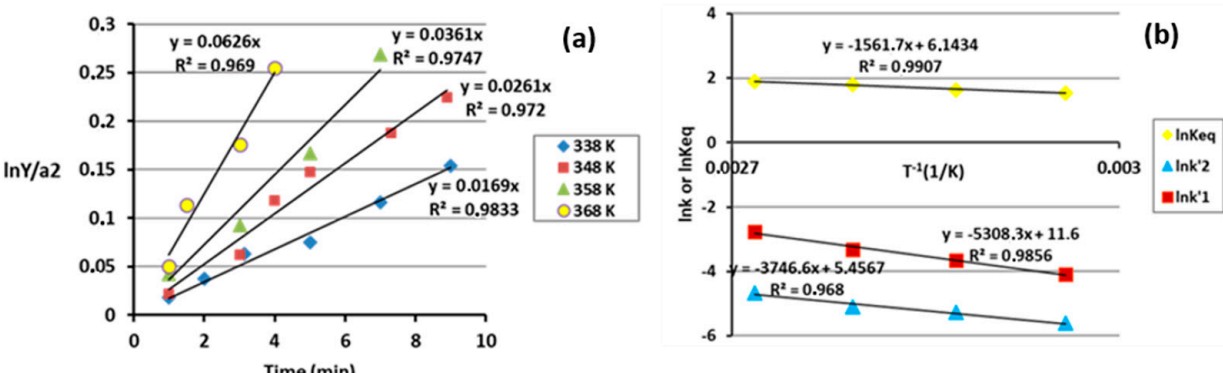

**Figure 2.** (**a**) Evaluation of rate constants of forward reactions $k_f$ at different temperatures (338–368 K) for the esterification reactions of levulinic acid with 1-hexanol; (**b**) Arrhenius and Van 't Hoff plots for the determination of $E_a$, $E_a^{-1}$, $\Delta H^0$, $\Delta S^0$ values.

Tables 1 and 2 report the calculated values of the kinetic and thermodynamic parameters of the reactions.

**Table 1.** Rate constants ($k_f$, $k_r$) and equilibrium constants evaluated for the esterification reactions of levulinic acid with 1-hexanol at different temperatures (338–368 K).

|  | **338 K** | **348 K** | **358 K** | **368 K** |
|---|---|---|---|---|
| $K_{eq}$ | 4.6 | 5.1 | 6.0 | 6.7 |
| $k_f$ | 0.0169 | 0.0261 | 0.0361 | 0.0631 |
| $k_r$ | 0.0036 | 0.0051 | 0.0060 | 0.0094 |

**Table 2.** Kinetics ($E_a$, $E_a^{-1}$) and thermodynamic parameters ($\Delta H^0$, $\Delta S^0$) evaluated for the esterification reactions of levulinic acid with 1-hexanol.

| $E_a$ **(KJ/mol)** | $E_a^{-1}$ **(KJ/mol)** | $\Delta H^0$ | $\Delta S^0$ |
|---|---|---|---|
| 43.3 | 29.6 | 13.7 | 53.3 |

For each temperature investigated, at the end of the reaction, two different phases were distinguished, which were then physically separated, weighted, and characterized (Table 3).

**Table 3.** Reaction systems obtained after the esterification of levulinic acid with 1-hexanol performed at four different temperatures (338–368 K) by using a 1:1:0.01 acid:alcohol:catalyst molar ratio.

| Temperature (K) | Conversion (%) | Phase Distribution (% wt) | | Phase Composition (%) | | | | | | | | | |
|---|---|---|---|---|---|---|---|---|---|---|---|---|---|
| | | | | Top Phase | | | | | Bottom Phase | | | | |
| | | Top | Bottom | A | H | C | W | L | A | H | C | W | L |
| 338 | 68.3 ± 0.4 | 96.8 | 3.2 | 23.8 | 14.2 | - | 2.4 | 59.6 | - | - | 34.2 | 65.8 | - |
| 348 | 69.4 ± 0.4 | 96.8 | 3.2 | 21.6 | 13.6 | - | 2.4 | 62.4 | - | - | 34.2 | 65.8 | - |
| 358 | 71.1 ± 0.5 | 96.7 | 3.3 | 17.4 | 13.2 | - | 2.4 | 63.0 | - | - | 35.3 | 64.7 | - |
| 368 | 72.2 ± 0.5 | 96.5 | 3.5 | 15.7 | 12.6 | - | 2.8 | 69.0 | - | - | 34.3 | 65.5 | - |

A—levulinic acid; H—1-hexanol; C—catalyst; W—water; L—hexyl levulinate.

### 3.2. Effect of the Amounts of 1-Hexanol and AlCl$_3$·6H$_2$O on the Reaction Yield

Considering the highest temperature (368 K), other reaction conditions were also studied, namely the use of higher amounts of 1-hexanol, considering an alcohol: acid molar ratio equal to 1 and 2, and AlCl$_3$·6H$_2$O, up to 10% molar with respect to the acid (Table 4).

**Table 4.** Systems obtained after the esterification reaction of levulinic acid with 1-hexanol, performed at 368 K, by using different amounts of alcohol and catalyst. Reaction conditions, conversions at the equilibrium, phase separations, and their compositions were determined.

| R | Catalyst (% mol) | Conversion (%) | Phase Distribution (% wt) | | Phase Composition (%) | | | | | |
|---|---|---|---|---|---|---|---|---|---|---|
| | | | | | Top Phase | | | | Bottom Phase | |
| | | | Top | Bottom | A | H | W | L | C | W |
| 1 | 1 | 72.2 | 96.8 | 3.2 | 15.7 | 12.6 | 2.8 | 69.0 | 34.6 | 65.4 |
| 1 | 5 | 76.5 | 90.1 | 9.9 | 14.0 | 8.3 | 1.8 | 76.4 | 52.9 | 47.1 |
| 1 | 10 | 83.1 | 84.7 | 15.3 | 9.7 | 8.5 | 0.9 | 80.9 | 65.1 | 34.9 |
| 2 | 1 | 85.5 | 97.9 | 2.1 | 5.3 | 37.0 | 3.5 | 54.2 | 35.6 | 64.4 |
| 2 | 5 | 90.8 | 93.1 | 6.9 | 3.4 | 36.0 | 2.2 | 58.7 | 52.6 | 47.4 |
| 2 | 10 | 92.5 | 88.3 | 11.7 | 2.7 | 33.8 | 1.4 | 61.8 | 59.9 | 40.1 |

A—levulinic acid; H—1-hexanol; C—catalyst; W—water; L—hexyl levulinate.

In Figure 3, the final mixtures related to the different catalyst loads were reported.

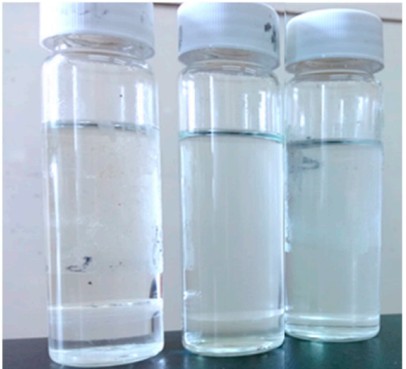

**Figure 3.** Biphasic system evaluated at the end of the esterification reaction of levulinic acid with 1-hexanol, carried out for 72 h, by using an alcohol: acid molar ratio equal to 1 and a catalyst loading equal to 1% (on the **right**), 5% (in the **middle**) and 10% (on the **left**) mol with respect to the acid.

Table 5 reports the conversion and Keq values evaluated when different alcohol and catalyst amounts were used, while Figure 4 shows the relationship between Keq and the catalyst amount.

**Table 5.** Reaction conditions and parameters of the esterification reactions of levulinic acid with 1-hexanol, performed at 368 K for 72 h.

| System | R | Catalyst (% mol) | χ Cat (mol cat/mol tot) | Conversion (%) | Keq |
|--------|---|------------------|-------------------------|----------------|-----|
| S1 | 1 | 1 | 0.0050 | 72.2 | 6.7 |
| S2 | 1 | 5 | 0.0164 | 76.5 | 14.9 |
| S3 | 1 | 10 | 0.0323 | 83.1 | 24.2 |
| S4 | 2 | 1 | 0.0050 | 85.5 | 4.4 |
| S5 | 2 | 5 | 0.0164 | 90.8 | 8.2 |
| S6 | 2 | 10 | 0.0323 | 92.5 | 11.5 |

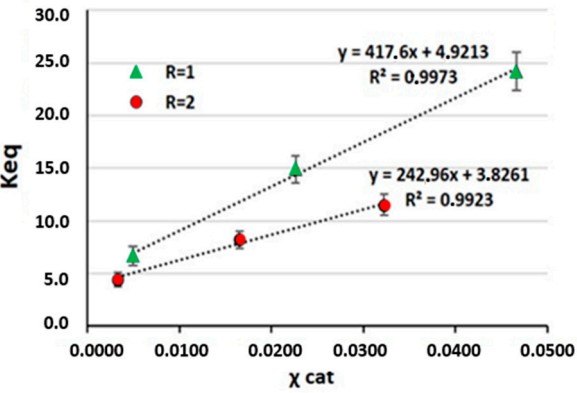

**Figure 4.** Relationship between χ cat and Keq for the esterification reaction of levulinic acid with 1-hexanol, performed at 368 K, for 72 h, by using $AlCl_3 \cdot 6H_2O$ as catalyst.

### 3.3. Catalyst Recoverability and Reusability

Catalyst recoverability and reusability were assessed; 1 g of the bottom phase obtained after the reaction of levulinic acid with 1-hexanol (levulinic acid: 1-hexanol: $AlCl_3 \cdot 6H_2O$ 1:2:0.1 molar ratio) was recovered by physical separation and subjected to thermal treatment (105 °C), in order to remove water and assess $AlCl_3 \cdot 6H_2O$ reusability. Weight loss of the lower phase, constituted by water (40.1 wt%) and $AlCl_3 \cdot 6H_2O$ (59.9 wt%), was monitored over time (0–120 h). In the first 7 h, water produced from the reaction was removed; from 7 to 120 h, the catalyst started to decompose, supposedly losing structural water and

hydrochloric acid, according to Reaction (1) [34], observing at the 120th hour, a weight loss equal to 49% wt of the catalyst starting weight.

$$2AlCl_3 \cdot 6H_2O \text{ (s)} \rightarrow Al_2O_3xHCl\ yH_2O \text{ (s)} + (6-x)HCl \text{ (g)} + (6-y)H_2O \text{ (g)} \qquad \text{(Reaction 1)}$$

Solid samples obtained after the thermal treatment performed for 7 h (100 wt% of $AlCl_3 \cdot 6H_2O$ starting weight) and 120 h (49% wt of $AlCl_3 \cdot 6H_2O$ starting weight) were tested as catalysts for the esterification reaction of levulinic acid with 1-hexanol. The sample obtained after a 7 h thermal treatment showed no reduction in the catalytic activity, allowing a conversion of levulinic acid equal to 92.1% after 5 h, not significantly different from that obtained by using a fresh catalyst (92.5 ± 1.0%), while the sample resulting from a 120 h thermal treating showed a high reduction in the catalytic activity, leading to a conversion of 39.0% after 24 h reaction. In Figure 5 are reported FT-IR spectra of fresh $AlCl_3 \cdot 6H_2O$ and $AlCl_3 \cdot 6H_2O$ recovered after 7 h and 120 h thermal treatment.

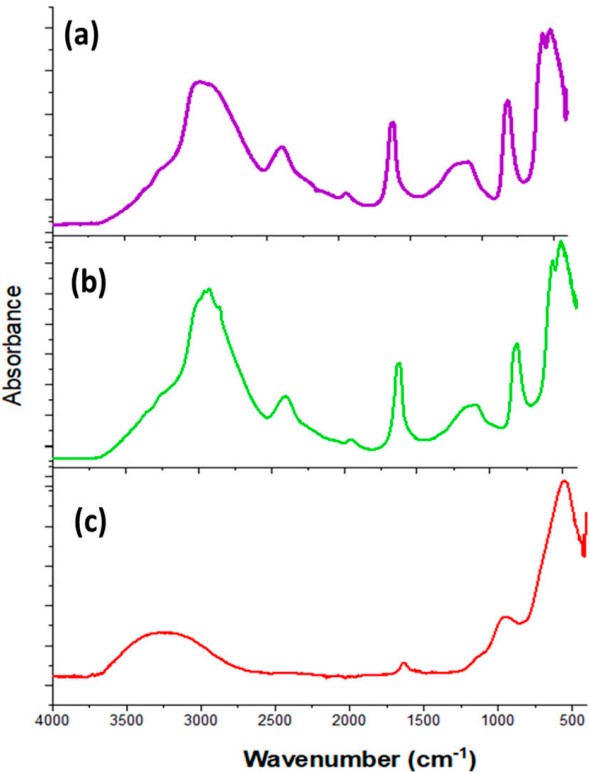

**Figure 5.** FT-IR spectra of (**a**) $AlCl_3 \cdot 6H_2O$, (**b**) solid catalyst recovered after 7 h thermal treatment of the bottom phase, (**c**) solid catalyst recovered after 120 h thermal treatment of the bottom phase.

Also, an under vacuum evaporation (bath oil: 75 °C, 30 mbar) of such phase was performed, until dryness of the catalyst, observing a weight loss equal to the amount of water initially present in such phase (40.1%, Table 4). The solid catalyst was then recovered and tested for a further reaction cycle, performed under the optimized reaction conditions, evaluating a conversion not significantly different from 92.5 ± 1.0%. This procedure was repeated three more times, also observing in these cases conversion values non-significantly different from 92.5 ± 1.0%.

### 3.4. Energy Demand for the Production and Purification of Hexyl Levulinate

A comparison between the energy required for hexyl levulinate synthesis and purification and the energy produced by hexyl levulinate combustion was performed. The energy demand for the synthesis of hexyl levulinate was calculated considering three main entries, namely the energy consumption due to the reaction carried out at 95 °C ($E_1$), water removal

for obtaining the solid catalyst to be reused ($E_2$), and the distillation process for purifying hexyl levulinate ($E_3$). $E_1$ was calculated as follows:

$$E_1 = (m_L \cdot Q_L + m_H \cdot Q_H)\Delta T \tag{6}$$

where $m_L$ and $m_H$ are the amounts (g) of levulinic acid and 1-hexanol, respectively, and $Q_L$ and $Q_H$ are the specific heat capacities for levulinic acid and 1-hexanol, equal to 2.11 J/(g·°C) [35] and 2.4 J/(g·°C) [36], respectively.

Energy consumption for water evaporation ($E_2$) in order to recover the solid catalyst was calculated as follows:

$$E_2 = m_W(Q_W \cdot \Delta T + \Delta H_{vap}) \tag{7}$$

where $m_W$ is the amount of water (g); $Q_W$ is water's specific heat capacity, equal to 4.18 J/(g·°C); $\Delta T$ is the temperature increase from 25 °C to 100 °C; and $\Delta H_{vap}$ is the enthalpy of vaporization of water, equal to 2.26 kJ/g. Lastly, energy consumption for 1-hexanol distillation from the upper phase in order to obtain a high-purity hexyl levulinate was determined (Equation (8))

$$E_3 = m_{Hf}(Q_H \cdot \Delta T + \Delta H_{vap}) \tag{8}$$

where $m_{Hf}$ is the amount of 1-hexanol (g) in the upper phase after the esterification reaction; $Q_H$ is the specific heat of 1-hexanol, equal to 2.4 J/(g·°C); $\Delta T$ the temperature increase from 25 °C to the boiling point of 1-hexanol, equal to 157 °C; and $\Delta H_{vap}$ is the enthalpy of vaporization of 1-hexanol, equal to 0.527 kJ/g [37].

Total energy consumption ($E_T$) can be expressed as the sum of the three entries (Equation (9)), and it was calculated as equal to 160.9 kJ.

$$E_T = E_1 + E_2 + E_3 = 44.3 + 36.8 + 79.8 = 160.9 \text{ kJ} \tag{9}$$

As for the energy produced from hexyl levulinate combustion, no data are available in the literature. An approximation was performed, considering the lower heating value of a shorter chain alkyl levulinate, namely n-butyl levulinate, for which the lower heating value (LHV) is equal to 28 MJ/kg [38]. The energy released in hexyl levulinate combustion was calculated as follows:

$$E_c = m \cdot LHV = 4468.8 \text{ kJ} \tag{10}$$

## 4. Discussion

A kinetic and thermodynamic study of the esterification reaction of levulinic acid with 1-hexanol, using $AlCl_3 \cdot 6H_2O$ as a catalyst, was performed under four different temperatures, namely 338, 348, 358, and 368 K. The maximum conversion was reached at 368 K, determining a conversion of 72.2 ± 0.5%, slightly higher than the lowest, determined at 338 K, equal to 68.3 ± 0.4%. In all these four cases, the most rapid increase in conversion was observed within the first 330 min, observing conversion values around 60%; obtaining higher conversions, up to the value determined at equilibrium at 72 h, instead required much longer times.

Kinetic and thermodynamic parameters were also calculated: the reaction was found to be slightly endothermic (Table 2), with a $\Delta H^0$ value equal to 13.7 KJ/mol, very similar to that determined for levulinic acid esterification with shorter chain alcohols (e.g., ethanol, butanol [19,39]). Also, $\Delta S^0$, equal to 53.3 J/(mol·K), was found to match these determinations, as well as Ea, assessed equal to 43.4 KJ/mol.

At the end of each esterification reaction, a biphasic system was observed: top phases consisted predominantly of hexyl levulinate, followed by the unreacted acid and alcohol, and a small amount of water; bottom phases were constituted of $AlCl_3 \cdot 6H_2O$, entirely distributed in such phases, and most of the water produced from the reaction (more than 90% wt) (Tables 3 and 4). No evidence of other compounds, namely levulinic acid, 1-

hexanol, and hexyl levulinate, was found in the bottom phases, even if trace amounts below the detection limits of the techniques used (<0.1% wt of the sample) could be present.

By using higher alcohol and catalyst amounts, a great improvement in levulinic acid conversion was achieved, obtaining, in the best case (acid:alcohol:AlCl$_3$·6H$_2$O 1:2:0.1 molar ratio), a conversion of 92.5% (Table 4). Increasing the AlCl$_3$·6H$_2$O amount, under the same alcohol–acid molar ratio, an increment in the bottom phase and a lower water amount in the top phase was evaluated due to the higher reaction yields (indeed, water is produced in larger amounts, and it is hardly miscible with the top phase when the amount of the ester increases), and also to the higher catalyst amount, entirely distributed in the bottom phase, which tends to trap water in that phase (Figure 3). To fully confirm this hypothesis, the upper phase obtained after having performed the reaction at 95 °C, with an acid: alcohol: catalyst 1:2:0.1 molar ratio, was separated from the lower phase, consisting of water and the catalyst (40.1% wt and 59.9% wt, respectively, Table 4) and quantified. To the upper phase, the same amount of water originally present in the lower phase was added in order to verify and quantify phase separation without the presence of AlCl$_3$·6H$_2$O. A phase separation was observed again, and it was quantified by removing water in the bottom phase. A total of 75% wt of the starting added water amount was insoluble, while the remaining part was solubilized in the upper phase, thus confirming the AlCl$_3$·6H$_2$O, being a salt, is able to decrease the solubility of water in the top phase.

The capability of AlCl$_3$·6H$_2$O to trap water in the bottom phase could be attributed to the increase in the conversion and, therefore, in the Keq value observed with the increase in AlCl$_3$·6H$_2$O amounts (Table 5). In detail, a direct proportionality between Xcat and Keq was detected (Figure 4), with a different equation line according to the amount of alcohol used (R = 1 and R = 2). The relationship between Keq and Xcat could be justified by the fact that the increase in the amount of AlCl$_3$·6H$_2$O decreases the amount of water that can actually hydrolyze the ester group (water in the top phase in the same phase of the ester), shifting to the right the reaction. As for the different equations for different amounts of 1-hexanol used (R = 1 and R = 2), this may be due to the amount of water in the top phase, which is greater, for the same amount of catalyst when a higher amount of alcohol is used.

According to a previous study, the homogenous catalysis of AlCl$_3$·6H$_2$O can be attributed to its solubilisation into systems containing water and/or alcohol. Several Al$^{3+}$ aquo-(alcohol) complexes were found to be present in such systems, which were the precursor of hydronium species that most presumably promotes the direct esterification as a very strong Brønsted acid [40].

Catalyst recoverability and reusability were also checked. The bottom phase obtained at the end of the esterification reaction (levulinic acid:1-hexanol:AlCl$_3$·6H$_2$O 1:2:0.1 molar ratio) was subjected to both a thermal treatment and a vacuum treatment to obtain the solid catalyst to be reused. When a thermal treatment of the bottom phase at 105 °C was performed, in the first 7 h, the evaporation of the water produced from the reaction was detected, obtaining a solid catalyst with the same composition as the starting catalyst (Figure 5), as well as no changes in the effectiveness in the catalysis of the esterification reaction. After 7 h, a decomposition of the catalyst started to occur, and an increasing weight loss over time was determined. The catalyst decomposed, presumably liberating HCl and H$_2$O, thus obtaining, after 120 h thermal treatment, a mixed catalyst which, unlike the fresh catalyst, was no longer completely soluble in aqueous systems and with a lower effectiveness in the catalysis of the esterification reaction. Indeed, in such a case, a lower levulinic acid conversion was determined, equal to 39.0%, much lower than that obtained with the fresh catalyst, equal to 92.1%. For such a catalyst, the FT-IR spectrum was acquired, showing the absence of some bands originally present in the aluminum chloride hexahydrate and a considerable reduction in the intensity of the bands that were still present (Figure 5). Also, Cl$^-$ and Al$^{3+}$ determination was performed, observing a stoichiometric ratio that was no longer the same as the starting catalyst (Al:Cl 1:3 stoichiometric ratio) but lower, approximately 1:1, further confirming the change in the chemical composition of the catalyst.

On the other hand, when a vacuum treatment was performed on the bottom phase for removing water, and the obtained solid catalyst was tested for a further reaction cycle, no significant difference from the initial levulinic acid conversion, equal to 92.5%, was observed. Moreover, this procedure was repeated four more times, obtaining the same levulinic acid conversion, confirming that $AlCl_3 \cdot 6H_2O$ fully maintains its catalytic effectiveness even after several reaction cycles. These results are in agreement with the elemental analysis (aluminum and chloride contents) performed on the same solid catalyst recovered after several reaction cycles, which confirmed that the amount of $Al^{3+}$ and $Cl^-$ was the same as that originally present in fresh $AlCl_3 \cdot 6H_2O$ [31]. If any changes would have occurred in the catalyst's chemical composition, for example, liberating corrosive HCl after vacuum evaporation, a residue compound having an unbalanced analysis (richer in aluminum and poorer in chlorides) should have been obtained.

$AlCl_3 \cdot 6H_2O$, therefore, joins together the advantages of homogeneous catalysis, namely the high catalytic activity, allowing the obtainment of very high conversions (92.5%), with those of heterogeneous catalysis, namely the possibility to be recovered and reused. Considering the possible energetic application of hexyl levulinate, it does make sense to perform a comparison between the energy consumption required for obtaining hexyl levulinate (as a pure product) and the energy that such compound is able to produce by combustion in order to have a preliminary idea of the feasibility of the process. Three main entries, presumably the most energy-intensive, were considered, namely the energy consumption due to the reaction carried out at 95 °C ($E_1$), water removal for obtaining the solid catalyst to be reused ($E_2$), and the distillation of 1-hexanol for purifying hexyl levulinate ($E_3$). If the best case is considered, namely a levulinic acid conversion equal to 92.5%, obtained by using an acid: alcohol: catalyst 1:2:0.1 molar ratio, considering a starting amount of levulinic acid equal to 100.0 g (and therefore 176.0 g of 1-hexanol), at the end of the reaction the following amounts are obtained: 7.5 g of levulinic acid; 94.6 g of 1-hexanol; 159.6 g of hexyl levulinate; 14.3 g of water. If the energy consumption due to the reaction carried out at 95 °C is considered ($E_1$), this could be approximated to the energy necessary to heat the initial mass of 100.0 g of levulinic acid and 176.0 g of 1-hexanol from room temperature to 95 °C (Equation (6)). $E_2$ was calculated by considering the evaporation of 14.3 g of water produced after the esterification reaction to obtain the solid catalyst. As for $E_3$, the energy consumption for the distillation of residual 1-hexanol in the upper phase (94.6 g) was considered. Total energy consumption was then determined, calculating a consumption of 160.9 kJ, of which $E_3$ was found to be the most important contribution (Equation (9)). As can be seen from Equations (9) and (10), the energy obtained from the combustion of hexyl levulinate (4468.8 kJ) is more than 27 times higher than the energy consumed for its synthesis and purification (160.9 kJ). Even if this represents a good result, a further increase could be made by reducing the $E_3$ contribution, deriving from the distillation of 1-hexanol. If an acid: alcohol molar ratio equal to 1:1 (with a catalyst amount of 10% molar with respect to the acid) is considered, even if at the expense of the reaction yield, there would be a much lower amount of 1-hexanol to be distilled. Using such a molar ratio, the conversion of levulinic acid, performing the reaction at 95 °C for 8 h, would be 83.1% (Table 4), and there would be a residual quantity of 1-hexanol equal to 14.9 g and, therefore, a noticeable reduction in $E_3$ value, equal to 12.6 kJ. Also, a slight reduction in $E_1$ and $E_2$ values would be determined, with $E_1$ equal to 29.6 kJ and $E_2$ equal to 33.2 kJ, and obviously, a reduction in the combustion energy of hexyl levulinate, being hexyl levulinate produced in a slightly lower amount, equal to 4004 kJ. In this case, considering that $E_T$ would be equal to 75.4 kJ, the energy of combustion of hexyl levulinate would be 53 times higher than the energy consumption for its synthesis and purification, which is a much better result than the one previously evaluated, considering an acid–alcohol molar ratio equal to 2. Obviously, it must be taken into account that a higher amount of unreacted levulinic acid would remain in the upper phase, and there would be the need to neutralize it, but, luckily, its sodium salt is not waste but an ingredient to be used as it is, for example in cosmetic formulations [41].

The process proposed for hexyl levulinate synthesis and purification (schematized in Figure 6) represents an efficient and clean process, and there are good prerequisites for its actual feasibility. The catalyst used, $AlCl_3 \cdot 6H_2O$, is a cheap, stable compound, it is less corrosive than the most commonly used mineral acids [19], and it could be obtained from Al-containing wastes after treatment of such wastes using HCl [42–44]. The utilization of such a catalyst allows the achievement of high reaction yields under mild reaction conditions, also ensuring a simplification of the product purification (being it already separated from the catalyst and from most of the water produced from the reaction), thus leading to reduced energy consumption; furthermore, it was found to be completely recoverable and reusable. All these aspects, in addition to the continuous research to obtain a low-cost levulinic acid and 1-hexanol from different types of biomasses, give good hope for the feasible, sustainable production of hexyl levulinate using $AlCl_3 \cdot 6H_2O$ as a catalyst.

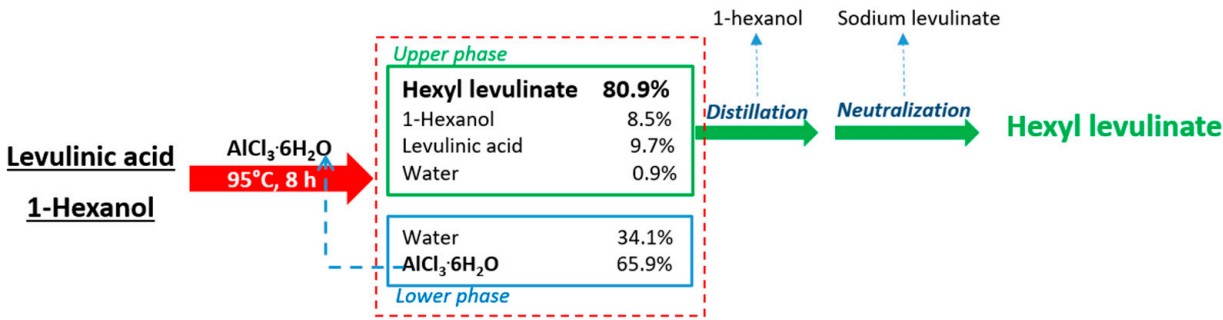

**Figure 6.** Schematization of hexyl levulinate synthesis and purification processes.

## 5. Conclusions

$AlCl_3 \cdot 6H_2O$ was tested as a catalyst for the esterification reaction of levulinic acid with 1-hexanol. Very high conversions, up to a value of 92.5% (acid:alcohol:catalyst 1:2:0.1 molar ratio), were obtained under mild reaction conditions, thanks to the high catalytic activity of $AlCl_3 \cdot 6H_2O$ and its ability to trap water, shifting to the right the equilibrium of the reaction. The product purification was simplified, being such a catalyst that is able to trap most of the water in the bottom phase, where it is completely distributed, well separated from the top phase, and mostly composed of hexyl levulinate, and the catalyst resulted as completely recoverable and reusable for several reaction cycles, without losing its catalytic effectiveness. From a preliminary assessment of energy consumption for hexyl levulinate synthesis and purification, performing the esterification reaction at 95 °C, using an acid: alcohol: catalyst 1:1:0.1 molar ratio, a reduced energy consumption was determined, which was 53 times lower than the energy eventually obtainable from hexyl levulinate combustion. The use of $AlCl_3 \cdot 6H_2O$ could therefore represent a promising efficient and clean route for obtaining hexyl levulinate.

**Author Contributions:** Conceptualization, V.D. and C.P.; methodology, V.D. and C.P.; validation, V.D. and C.P.; formal analysis, V.D. and C.P.; investigation, V.D.; resources, C.P.; data curation, V.D. and C.P.; writing—original draft preparation, V.D. and C.P.; writing—review and editing, V.D. and C.P.; visualization, V.D. and C.P.; supervision, C.P.; project administration, C.P.; funding acquisition, C.P. All authors have read and agreed to the published version of the manuscript.

**Funding:** This research was funded by MIUR, grant number Code: PE8-11775-2017FWC3WC_004 PRIN-VISION Project.

**Institutional Review Board Statement:** Not applicable.

**Informed Consent Statement:** Not applicable.

**Data Availability Statement:** Data sharing not applicable.

**Conflicts of Interest:** The authors declare no conflict of interest.

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
