# Peer review of "Effective Upgrading of Levulinic Acid into Hexyl Levulinate Using AlCl3·6H2O as a Catalyst"

_2673-8783, doi:10.3390/biomass3030016_

Round 1
Reviewer 1 Report
This manuscript focuses on the synthesis of long-chain levulinate ester, hexyl levulinate as the example. Long-chain levulinate esters have a series of merits and are a class of potential fuel chemicals. The investigation on the preparation of long-chain levulinate ester is significant. The manuscript can be published, while some comments and questions should be carefully addressed prior to the acceptance.
Comments:
1. What does “remove water” mean? How does the catalyst remove the water?
2. Since water is produced by the esterification, does AlCl3 react with water?
3. What is the actual catalyst for the conversion? AlCl3 or HCl or Al3+ cation?
4. Does other aluminum salts such as sulfate and nitrate work for the esterification?
5. I suggest the authors to give the yield or selectivity data but not the phase composition. The data will facilitate understanding how about the reaction performance and side reaction.
Reviewer 2 Report
The manuscript of D’Ambrosio et al., investigated the production of hexyl levulinate through AlCl3.6H2O. Although alkyl levulinate production is studied already (with numerous types of homogeneous/heterogeneous catalysts), there are some interesting aspects of this study.Therefore this paper deserves recognition in this field of research. The approach of the authors is well performed, the article is well written and the applied methodology merits publication in Biomass. However, after reading the manuscript I have some minor and major comments which ought to be addressed before publication:
- - Line 71, “cycleS”.
- - Please specify exactly which reactant/product is characterized by what technique. The reviewer assumes that levunic acid and hexyl levulinate is determined via GC-FID. But if levulinic acid is already determined via GC-FID, why bother determining its concentration via titration? Is there a discrepancy between both analyses?
- - It might also be interesting to add the limit of detection of the techniques applied. The authors mention for instance that no levulinic acid is detected in the bottom phase (where water is present). Not a single trace of levulinic would be strange, especially as there is still levulinic acid present in the top phase. Normally there should be a partitioning of levulinic acid between both phases. (The example is made for levulinic acid, but not limited to levulinic acid).
- - How did the authors separate the 2 phases? Via (micro)pipetting, separating funnel, … ? Please specify. This could influence potential standard errors.
- - It is difficult for the reader to see the differences in reaction temperature based on Figure 1. The authors should think of a better way of presenting the data. The most important data is probably in shorter reaction times and between conversions of 60 - 80%. Now all the data points are just clustered making it visually difficult to assess the different data points (and therefore temperature effects).
- - Typo line 140, 144, 148 “Vant’Hoff’ should be “Van’t Hoff”.
- - Please enlarge Figure 2 and make sure the text does not overlap axes and/or lines.
- - Did the authors obtain solid AlCl3.6H2O after evaporation for the catalyst reusability experiment?
- - The authors make the following statement: “AlCl3 .6H2O therefore joins together the advantages of homogeneous catalysis, namely the high catalytic activity, allowing the obtainment of very high conversions (92.5%), with those of heterogeneous catalysis, namely the possibility to recover and reuse the catalyst. In this way an efficient and clean process could be fulfilled, reducing the use of resource and the generation of wastes, if compared to the synthesis performed by using mineral acids” If the recovery process is based on the evaporation of (mainly) water, there should be some nuance here. Namely, it would cost considerable amounts of energy to recycle and reuse it.
See comments above.
Round 2
Reviewer 1 Report
The revisions and answers are fine to me. I think that the authors have carefully addressed the previous comments. It can be accepted.
Reviewer 2 Report
The authors addressed the comments raised by the reviewer and therefore the reviewer accepts this manuscript for publication in Biomass.